# Films Based on Biopolymers Incorporated with Active Compounds Encapsulated in Emulsions: Properties and Potential Applications—A Review

**DOI:** 10.3390/foods12193602

**Published:** 2023-09-28

**Authors:** Camily Aparecida Reis, Andresa Gomes, Paulo José do Amaral Sobral

**Affiliations:** 1Department of Food Engineering, Faculty of Animal Science and Food Engineering, University of São Paulo, Pirassununga 13635-900, SP, Brazil; camilyreis@usp.br (C.A.R.); pjsobral@usp.br (P.J.d.A.S.); 2Food Research Center (FoRC), University of São Paulo, Rua do Lago, 250, Semi-Industrial Building, Block C, São Paulo 05508-080, SP, Brazil

**Keywords:** active films, proteins, polysaccharides, lipophilic compounds

## Abstract

The rising consumer demand for safer, healthier, and fresher-like food has led to the emergence of new concepts in food packaging. In addition, the growing concern about environmental issues has increased the search for materials derived from non-petroleum sources and biodegradable options. Thus, active films based on biopolymers loaded with natural active compounds have great potential to be used as food packaging. However, several lipophilic active compounds are difficult to incorporate into aqueous film-forming solutions based on polysaccharides or proteins, and the hydrophilic active compounds require protection against oxidation. One way to incorporate these active compounds into film matrices is to encapsulate them in emulsions, such as microemulsions, nanoemulsions, Pickering emulsions, or double emulsions. However, emulsion characteristics can influence the properties of active films, such as mechanical, barrier, and optical properties. This review addresses the advantages of using emulsions to encapsulate active compounds before their incorporation into biopolymeric matrices, the main characteristics of these emulsions (emulsion type, droplet size, and emulsifier nature), and their influence on active film properties. Furthermore, we review the recent applications of the emulsion-charged active films in food systems.

## 1. Introduction

Food packaging is used to maintain the safety and quality of food products during storage and transportation, protecting them against undesirable external factors. Consumer demand for convenient, safer, healthier, and fresher-like food with improved shelf life and environmental concerns have given new concepts and functions to food packaging [1]. In addition to the usual inert passive containment and protective properties, these novel food packaging materials should have active compounds to enhance the performance of the packaging system and should preferably be produced with biodegradable ingredients from renewable sources [2,3,4]. As a result, there is a growing demand for biopolymer-based active packaging, which has stimulated studies searching for new strategies, boosting the development of a novel generation of food packaging.

Active packaging systems are produced by incorporating active ingredients into the packaging material or the package’s headspace, which can release or absorb substances into or from the packaged food or from the environment surrounding the food, extending its shelf life without compromising its quality and safety [5]. These active packaging systems can perform diverse functions: (i) absorbing/scavenging substances such as carbon dioxide, oxygen, ethylene, and moisture; (ii) releasing/emitting compounds, such as ethanol, antioxidants, preservatives, flavors; (iii) food component removal (e.g., lactose, cholesterol); and (iv) microbial and quality control [6].

Active films based on biopolymers are promising materials for active packaging design. In addition to the functional properties promoted by the incorporation of active compounds, such as antimicrobial and antioxidant compounds, these film types are an alternative to synthetic packaging materials as they are produced from biodegradable, renewable, and environmentally friendly materials [7,8]. The most common biopolymers used in the production of active films are polysaccharides (e.g., starch, chitosan, pectin, cellulose derivatives, gums) and proteins (e.g., gelatin, whey protein, caseinate, soy protein, zein) [9].

Regarding functionality, a wide range of active compounds have been added to the biopolymeric matrix of active films [6,10], such as extracts from herbs and spices [11,12] or from agricultural by-products [13,14], essential oils (EO) [15,16], and lipophilic compounds [17,18,19]. Antimicrobial and antioxidant agents are the most studied active components since the growth of pathogenic and/or spoilage microorganisms and oxidative degradation are the main causes of food spoilage [6].

In recent decades, interest in adding lipophilic bioactive components in films has increased, as they are powerful antimicrobial and antioxidant agents [20,21]. There are several studies directly incorporating lipophilic compounds into filmogenic matrices; however, the chemical and physical stability of these films is limited [22,23]. Furthermore, these bioactive compounds have low solubility in water due their high hydrophobicity and are susceptible to oxygen, temperature, pH, and ionic strength. These characteristics hinder the incorporation of hydrophobic functional compounds in hydrophilic matrices and can reduce their bioavailability [24,25].

One strategy to overcome these challenges is to encapsulate lipophilic bioactive compounds in the oil phase of oil-in-water (O/W) emulsions prior to their incorporation into a filmogenic solution. The use of O/W emulsions allows the dispersion of a hydrophobic bioactive compound and an oil phase in a hydrophilic matrix [17,19,26,27,28]. Water-in-oil-in-water (W/O/W) emulsions can also be used to encapsulate hydrophilic compounds before their addition into biopolymeric matrices [29,30,31]. The emulsions can protect active ingredients against external factors and promote their controlled release [32,33,34,35]. The bioactive compounds can be encapsulated by using several techniques, producing solid lipid nanoparticles (SLN), liposomes, and nanostructured lipid carriers (NLC), structured oils, and nanoparticles, among others [36]. Nevertheless, emulsions are produced more easily than these others products, and they more easily incorporated into aqueous systems [37].

In this context, the aim of this review is to present and critically discuss how emulsions encapsulating bioactive compounds can affect the properties of films incorporated into these systems. Active compound retention, active film stability, and release behavior in food simulants are also pointed out. The first part is dedicated to the main types and characteristics of emulsions encapsulating active compounds. The second part is a review of the main studies on the active films incorporated with emulsions encapsulating active compounds and a comparison between the incorporation of active compounds in their native form or when emulsified. The third part discusses the influence of emulsion characteristics on the film properties. In the last part, this review presents some possible applications of these active films in food systems.

## 2. Emulsions

Emulsions are colloidal systems consisting of two or more completely or partially immiscible phases, such as oil and water. For a liquid system, one of the liquids is dispersed as droplets (dispersed phase) in another liquid continuous phase forming the single oil-in-water (O/W) or water-in-oil (W/O) emulsion [38]. More complex systems, such as water-in-oil-in-water (W/O/W) or oil-in-water-in-oil (O/W/O) emulsions, can also be produced from single emulsions. In a W/O/W double emulsion, smaller water droplets are dispersed in an oil phase (W/O) [39,40], which is subsequently dispersed in an external aqueous phase (W) [41]. The double emulsions can concurrently encapsulate hydrophilic and hydrophobic bioactive molecules; however, the stabilization of two different interfaces during the emulsion processing and storage is still a challenge [42,43,44]. Figure 1 shows a schematic view of the main emulsion designs.

The emulsions are thermodynamically unstable and tend to separate into an oil and an aqueous phase because the free energy of the separated phases is lower than the emulsified system [45]. However, a great kinetic stability can be reached by adding stabilizers, such as emulsifiers (surfactants, proteins, and carbohydrates) or solid particles [37,46]. These compounds can improve the stability of systems over time by retarding or preventing destabilization mechanisms such as creaming/sedimentation, flocculation, coalescence, and Ostwald ripening [47]. Unlike traditional emulsions, microemulsions are thermodynamically stable systems composed of nanometer-sized particles (formed by oil, surfactant/co-surfactant) dispersed in an aqueous phase [48] (Figure 2). Readers interested in emulsion stabilization fundamentals are invited to read the excellent book by McClements [38].

Emulsions can be classified according to droplet size in emulsions or nanoemulsions. Usually, nanoemulsions present droplets ranging between 20 and 200 nm, while colloidal dispersions with droplet diameters larger than 200 nm are named emulsions [45,49]. However, different average droplet diameters are frequently used as the demarcation point between nanoemulsions and emulsions, which can range between 20 and 200 nm [49,50,51]. Although both colloidal systems are thermodynamically unstable, a reduction in droplet size can enhance the kinetic stability and increase the specific surface area, improving the bioactivity of lipophilic active compounds vehiculated in nanoemulsions [49].

Stabilization by solid particles is another approach to improve the kinetic stability of emulsions, which are called Pickering emulsions. Compared with conventional emulsions, Pickering emulsions can be stable for a longer period as micro- or nanometer-sized particles irreversibly adsorb at interfaces [52]. Furthermore, this type of stabilization offers distinct advantages, especially in the field of delivery of bioactive compounds, such as good long-term kinetic stability, low cytotoxicity, controlled release under specific conditions (such as temperature, pH, light intensity, ionic strength), and targeting of the bioactive compound for enhanced functionality in food [53,54,55,56].

Overall, emulsified systems have been used to encapsulate various hydrophobic active compounds, such as hesperidin [24,57], curcumin [58,59], resveratrol [60], β-carotene [61], pepper oil [62], and a wide range of EOs (e.g., oregano, lemongrass, cinnamon, clove, American mint, and pectinata) [63,64,65], among others. The use of emulsions as an encapsulating system for bioactive compounds can prevent their degradation, preserve their bioactivity, and improve their performance in different ways: (i) protecting the bioactive compounds from pro-oxidant molecules and environmental conditions that could degrade them via the continuous phase and the interfacial layer of the emulsion [59,66]; (ii) allowing the incorporation and uniform distribution of hydrophobic molecules and the oil phase in hydrophilic biopolymeric matrices [67,68]; (iii) masking undesirable flavors [36,69]; and (iv) promoting the controlled release of active compounds [70,71].

Because of these advantages, emulsified systems such as Pickering emulsions [72,73,74], nanoemulsions [19,68,75], and emulsions [76,77] have been widely used to incorporate hydrophobic compounds into biopolymeric matrices. Although most active compounds encapsulated in O/W emulsions and incorporated into a biopolymer matrix are hydrophobic, recently, double emulsions (W/O/W) charged with hydrophilic active compounds in the inner aqueous phase were successfully used to produce active films [29,30]. A similar W/O/W system was also used to simultaneously encapsulate hydrophilic (nisin) and hydrophobic (carvacrol) active compounds and to fabricate chitosan-based films to preserve salmon fillets [31].

## 3. Films Incorporated with Bioactive-Compound-Charged Emulsions

Figure 3 shows a schematic representation of the production system of active films incorporated with emulsions. First, the emulsion is produced with the active compound in the oil phase. After that, the emulsion is added to a biopolymeric solution, and the mixture is carefully homogenized, forming a homogeneous film-forming solution (FFS). On a laboratory scale, films are usually produced by the casting method, in which the FFS is poured onto a plate support and dried until the solvent evaporates [78]. More options to fabricate films (moist and dry techniques) and their advantages and disadvantages can be found in the review by Mellinas, et al. [79].

When active films are applied as food packaging, they can offer protection against light, moisture, and gas migration from/to the external environment of the packaging. In addition, they can release active ingredients in the package’s internal space, promoting an antioxidant and/or antimicrobial effect that can extend the food product’s shelf life.

To the best of the author’s knowledge, the first works on the development of active films by the incorporation of encapsulated active compounds in nanoemulsions were carried out by Bilbao-Sáinz, et al. [81], who studied the incorporation of nanoemulsions protecting EOs in isolated soybean protein-based films [81], and by Otoni, et al. [82], who studied the incorporation of a nanoemulsion encapsulating cinnamaldehyde in pectin/papaya puree films [82]. Previously, some articles studying emulsified films were published; nevertheless, in these cases, the emulsifying process occurred in the film-forming solution. The oil phase (containing active compounds) and the emulsifier were directly added into the biopolymeric film-forming solution [83,84,85,86]. However, it is not the purpose of this review to approach this type of study.

In the last five years, the number of published articles on emulsion-incorporated active films has increased, considering all the emulsion types (nanoemulsions, emulsions, double emulsions, and Pickering emulsions (Figure 4). A search on the Web of Science Core Collection using “emulsion” and “active film” as the topic and selecting the “Food Science & Technology” category led to 161 results, of which 117 were published from 2018 [87]. When other terms such as “nanoemulsion”, “microemulsion”, “Pickering emulsion” or “double emulsion” were added to the search it was possible to note that this research area has a significant potential to grow since the influence of the emulsion (emulsion type, droplet size, and nature of the emulsifier or solid particle) on the film structure is not fully understood.

Recent studies on the encapsulation of lipophilic bioactive compounds into emulsions to design active films are presented in Table 1. The most commonly used biopolymers for producing active films loaded with bioactive compounds encapsulated in emulsions include starch from different sources, gelatin from various sources, and chitosan, among others, which are less common. Blends composed of two or more biopolymers have also been widely studied (Table 1).

Essential oils are the more commonly used bioactive compounds for developing biopolymer-based active films due to their natural origin and excellent antimicrobial and antioxidant properties [10]. Among the EOs, cinnamon and clove EO are the mainly studied compounds (Table 1).

### Emulsion-Encapsulated Bioactive Compound versus Bioactive Compound in Native Form

Some authors have evaluated the advantages of producing active films with emulsified active compounds [22,23,119]. One way to assess this is by comparing the addition of the native form of the active compound versus its emulsified form.

The comparison of adding bergamot oil directly into the WPI matrix and in a nanoemulsified form was evaluated by Sogut (2020). The films with bergamot oil nanoemulsions stabilized by nanocellulose presented a higher tensile strength and elastic modulus, and a lower elongation at break than those directly incorporated with bergamot oil. Moreover, nanoemulsion-charged films had lower a WVP than systems incorporated with bergamot oil at the same oil concentration. The addition of nanodroplets increased the tortuosity of films, consequently reducing the mass diffusivity. The nanoemulsions decreased the UV-Vis light transmittance and increased the opacity of the films compared with control (pure WPI film) and direct incorporation oil systems. These results suggest that films with a nanoemulsion can be used as a light barrier. In addition, the presence of nanoemulsions stabilized by nanocellulose delayed the release of the bioactive compound in food simulant fluid (ethanol 50%). This behavior was associated with the closer structure formed by the interaction between WPI and bergamot oil in the presence of nanocellulose [23].

Curcumin-loaded Pickering emulsions and curcumin solution were incorporated into corn starch/PVA blend films to produce intelligent pH indicator films [22]. Curcumin-solution film was more stretchable and resistant compared with control and Pickering emulsion systems, and a considerable decrease in tensile strength and elongation at break was observed in Pickering-emulsion films. Moreover, WVP and oxygen permeability decreased in films with the presence of Pickering emulsions. Regarding functionality, active films with curcumin solution showed lower antioxidant activity than those with curcumin protected by Pickering emulsions. During the production process of the films, the curcumin added directly into the biopolymeric matrix was more easily degraded when exposed to heating, cavitation, and drying. On the other hand, the thick interfacial layer of Pickering emulsions protected the curcumin from oxidation during the production of the films, preserving its bioactivity. Furthermore, Pickering emulsions also reduced the rate of curcumin degradation by light during storage, while curcumin-loaded films showed a rapid decrease in antioxidant activity over time. For the same reason, the emulsified films presented higher antimicrobial activity than the free-curcumin-loaded one [22].

Xanthan-gum-based FFSs containing EOs (clove, cinnamon, and oregano) and nanoemulsified EOs were evaluated in relation to their in vitro antimicrobial activity [119]. At the same concentrations of EOs, the active FFS with free active compounds was more effective in reducing microbial growth than the FFS with nanoemulsified oils. This behavior may be associated with components such as Crodamol^TM^ (saturated triacylglycerol) and surfactants, which form additional barriers to the diffusion the EOs, controlling their release [119]. It is important to point out that the active compounds were not heated during the FFS production and were not subjected to drying. Thus, the protective effect of emulsions from heating and during storage was not evaluated in this case.

## 4. Influence of Emulsion Characteristics in Active Film Properties

There are many advantages to using emulsified bioactive compounds in producing active films, as seen in the previous section. However, the incorporation of emulsions affects the physicochemical and functional properties of films, such as their mechanical and barrier properties and antimicrobial and antioxidant activities. It is interesting to highlight that characteristics of the emulsified systems, such as emulsion type, droplet size, and emulsifier nature, can also influence the structure and functionality of active films, as well as their performance as active packaging, as discussed in more detail below (Figure 5) [26,72,95,96].

### 4.1. Effect of Emulsion Type

Different emulsions can be designed, like emulsions, nanoemulsions, Pickering, and double emulsions, using different homogenization techniques and stabilizers. The obtained systems show distinct structures, stabilization mechanisms, and kinetic stabilities. These characteristics can influence the properties of active films charged with emulsions, such as morphology, moisture, water vapor permeability, and mechanical properties. Several factors, such as the biopolymer and plasticizer type and their concentration or even emulsion concentration, can also affect film properties. However, Table 2 summarizes only the main properties influenced by the emulsion type, droplet size, and emulsifier nature as examples.

Liu, et al. [72] studied chitosan films loaded with an O/W emulsion or Pickering emulsions carrying the cinnamon EO. Films loaded with the O/W emulsion presented lower water vapor permeability (WVP) than the control films; however, Pickering emulsions increased the WVP compared with control and O/W emulsion-charged films [72]. In contrast, nanoemulsions and Pickering emulsions encapsulating clove EO reduced the WVP of pullulan/gelatin-based films, but the lowest values were observed in films containing Pickering emulsions [80]. Similar behavior was observed in pectin films with added nanoemulsions or Pickering emulsions carrying marjoram (*Origanum majorana* L.) EO. The decrease in the WVP was related to the hydrophobic nature of the system due to the oil-phase presence; however, the Pickering emulsion was more effective than nanoemulsions in reducing the WVP. Both emulsified systems increased the cross-linking between pectin chains due to the filling of free spaces in the pectin matrix, which reduced the mobility of the pectin chains and, therefore, the migration rate of water vapor molecules [103]. Double emulsions (W/O/W) have recently also been applied in biopolymeric films. Double emulsions carrying the “Pitanga” leaf hydroethanolic extract reduced the WVP of chitosan, gelatin, and chitosan-gelatin blended films [30]. Double emulsion (W/O/W) allows the incorporation of multiple types of active compounds, such as hydrophilic (into the inner W phase) and hydrophobic (into the intermediary O phase) compounds. Moreover, they can be easily incorporated into a film-forming solution prepared using water as a solvent [75,76].

Different types of emulsions can affect other water-related properties. For example, the moisture content decreased with the addition of Pickering emulsions containing clove EO into pullulan/gelatin-based films, while nanoemulsions did not affect this property [80]. In contrast, moisture absorption increased with the incorporation of nanoemulsions but decreased when Pickering emulsions were added. This phenomenon may have occurred due to the interaction between emulsion droplets and the matrix, improving the waterproofing of the film [80]. The same behavior in moisture absorption was observed in pectin-based films charged with nanoemulsions and Pickering emulsions encapsulating marjoram EO. Authors also attributed this behavior to an interaction between oil droplets and the biopolymer chains in the matrix [103]. In contrast, a double emulsion encapsulating the “Pitanga” leaf extract did not affect the moisture content of chitosan and gelatin films; however, the solubility in water was reduced. The addition of the oil phase promoted a reduction in the hydrophilicity of the film matrix, resulting in decreased solubility [30]. Water contact angles increased with the incorporation of emulsions and Pickering emulsions compared with control film; however, no difference between both emulsion-charged films was observed, indicating that the higher contact angle can be attributed to an increase in oil content and the rougher surface of the films [72]. On the other hand, a double emulsion provoked a decrease in the water contact angle of gelatin and gelatin-chitosan blended films. Authors attributed this behavior to the non-polar substances present in the hydroethanolic extract [30]. In the same way, Pickering emulsions and nanoemulsions carrying clove EO affected the surface morphology of pullulan/gelatin-based films. Pickering emulsions improved the irregularity and the roughness of the film’s surface, and the nanoemulsion reduced the roughness, which was attributed to Tween 80 [80]. The double emulsion also improved the surface roughness of chitosan, gelatin, and chitosan-gelatin blended films [30].

Regarding mechanical properties, the tensile strength of Pickering emulsion-loaded chitosan films was higher than that incorporated with an O/W emulsion, in addition to presenting a greater elongation at break. The cellulose nanocrystals (CNC) used as stabilizers adsorbed on the oil–water interface of Pickering’s emulsions acted as reinforcement fillers for the films, mitigating the effect of breaking the continuous structure of the films promoted by the addition of cinnamon EO. However, both emulsion types reduced the tensile strength and elongation at break compared with the control film [72]. On the other hand, Pickering emulsions enhanced the tensile strength of pullulan/gelatin-based films. However, these films were less stretchable than control films and films with nanoemulsions. Nanoemulsions also caused a reduction in tensile strength and an increase in the elongation at the break of films compared with the control one [80]. Nanoemulsions also provoked a decrease in the Young’s modulus and the tensile strength in pectin films (except for the sample with 2.5% nanoemulsion), but nanoemulsions did not affect the elongation at break. In contrast, adding 5 and 7.5% of the Pickering emulsion increased the Young’s modulus and tensile strength of the films. However, the elongation at break decreased in the sample with 7.5% of the Pickering emulsion and increased in the sample with 5%, which showed higher elongation at break than the control sample [103]. When the double emulsion was added to the gelatin-based film, improved tensile strength and elongation at break were achieved; however, it did not affect the elongation at break of the chitosan film [30]. On the contrary, the incorporation of double emulsion (up to 1% *w*/*w*) increased the tensile strength and decreased the elongation at the break of chitosan films in relation to the control film. A continuous increase in the emulsion concentration (from 1% to 2.5% *w*/*w*) caused a reduction in tensile strength [31].

The incorporation of emulsions encapsulating cinnamon EO in chitosan films decreased the films’ transparency. However, Pickering emulsion-charged films showed lower transparency than that with a Tween-80-stabilized O/W emulsion. This behavior was justified by the use of CNCs as stabilizers in Pickering emulsions, which promoted higher light scattering [72]. The same behavior was observed in gelatin, chitosan, and gelatin–chitosan films charged with double emulsions [30]. Pullulan/gelatin-based films loaded with Pickering emulsions carrying clove EO exhibited a higher barrier to light than films loaded with nanoemulsions [80]. The presence of the Pickering emulsions decreased the light transmittance, probably due to a shadow effect of the stabilizing particles [17].

Active properties of films, such as antioxidant and antimicrobial activity, are also influenced by emulsion type. Pullulan/gelatin-based films incorporated with nanoemulsions carrying clove EO showed higher antioxidant activity, measured by DPPH (2,2-Diphenyl-1-picrylhydrazyl) and ABTS (2,2′-azino-bis(3-ethyl-benzothiazoline-6-sulfonic acid) radical scavenging, than those incorporated with Pickering ones. This fact can be linked to the lower release of EOs in Pickering emulsion-charged films than in nanoemulsion-loaded films. The WPI–inulin complex used to stabilize the Pickering emulsion, adsorbed at the oil–water interface, may have reduced the release of the active compound [80].

Also comparing the addition of a nanoemulsion and a Pickering emulsion encapsulating marjoram (*Origanum majorana* L.) EO in pectin-based films, Almasi, et al. [103] reported that antioxidant activity assessed by DPPH was higher in films containing nanoemulsions than in films containing Pickering ones. This phenomenon can be caused by the decreased mobility of the loaded active compounds through the matrix in the films incorporated with Pickering emulsions [103]. A double emulsion (W/O/W) encapsulating the “Pitanga” hydroethanolic leaf extract improved the antioxidant activity as evaluated by the ABTS, DPPH, and FCRC (Folin–Ciocalteu reducing capacity) of gelatin, chitosan, and chitosan–gelatin blended films. However, gelatin film presented the highest antioxidant activity compared with the other matrices, which could be explained by the higher protective effect of gelatin compared to the other biopolymers during processing [30].

In addition to emulsion type, active properties can be affected by the emulsion concentration in the biopolymeric matrix. For example, in pullulan/gelatin-based films, raising the concentration of the nanoemulsion and Pickering emulsions enhanced the ABTS and DPPH radical scavenging [80]. The same occurred in pectin films for both emulsion types [103]. Increasing the concentration of a double emulsion carrying carvacrol improved the antioxidant capacity of chitosan films, measured by the DPPH method [31]. This same behavior has been reported by several authors [73,94,99,108].

Another active property of interest is the antimicrobial activity. Chitosan films incorporated with emulsions encapsulating cinnamon EO did not present significant inhibition zones against *E. coli* and *S. aureus* compared with the control film. However, Pickering emulsion-loaded films improved the inhibition zone against these bacteria. These different behaviors were attributed to the limited release of the active compound vehiculated in emulsions [72]. On the contrary, there was no significant difference in the antibacterial activity of pectin-based films charged with Pickering emulsions and nanoemulsions carrying marjoram EO [103].

### 4.2. Effect of Emulsion Droplet Size

Emulsions can be produced with a wide range of droplet sizes, ranging from a few nanometers (<50 nm) to micrometers (>100 μm), depending upon the ingredients and the homogenization methods utilized to produce them [38,121]. Droplet size influences important emulsion properties, such as stability, optical properties, rheology, and release rate [48,122,123]. In addition, the droplet size of emulsions also affects the characteristics of active films charged with emulsions, as will be discussed below.

Different homogenization conditions changed the mean diameter of the droplets from 235.34–355.36 nm to 52.53–73.41 nm in licorice EO-loaded emulsions. The roughness increased when emulsions with larger droplets were incorporated into CMC films, while no effect on the roughness was observed in films with smaller droplets [26]. Otherwise, roughness parameters decreased in konjac glucomannan films with the increase of the mean droplet diameter of Pickering emulsions. Authors attributed this behavior to the smaller number of bigger droplets in the biopolymeric matrix [120].

Water-related properties, such as moisture content and water solubility, were not affected by droplet sizes in starch films [77]. Unlike in konjac glucomannan films, the moisture content and water solubility decreased with the larger droplet sizes (42.86 ± 15.27 μm) of Pickering emulsions. The different droplet sizes were obtained by changing the oil-phase amount (higher oil content resulted in larger droplets). Therefore, the effect of the larger droplet size can be related to the hydrophobicity of the oil droplets and interactions between them and the matrix, which could have partially replaced the matrix–water interactions [120]. Evaluating the water contact angle of the films’ surface, konjac glucomannan films charged with Pickering emulsions with larger droplets (42.86 ± 15.27 μm) presented higher values of water contact angle, which may be related to the increase of the oil phase in emulsions [120]. Nevertheless, droplet size did not affect the water contact angle of starch films with the same oil concentration added to carnauba-wax emulsions [77].

Comparing the influence of droplet size on the WVP of CMC films charged with emulsions encapsulating licorice EO, Fattahi and Seyedain-Ardabil [26] reported that emulsions with bigger droplets (235.34–355.36 nm) increased the WVP compared with the control film, while smaller ones (52.53–73.41 nm) provoked a decrease in the WVP [26]. Likewise, Oliveira Filho, et al. [77] observed a lower WVP with the addition of emulsions with nanometric droplets compared with micrometric droplets in starch-based films [77]. Zhao, et al. [88] also observed a reduction in WVP values when Pickering emulsions with smaller droplets were incorporated in chitosan/anthocyanidin films [88]. On the contrary, Liu, et al. [120] reported that the larger droplet sizes of Pickering emulsions decreased the WVP values of konjac glucomannan films.

Various emulsion droplet sizes caused a plasticizing effect in the CMC matrix. Emulsions with mean diameter droplets smaller than 100 nm produced films with lower tensile strength and higher elongation at break than those with bigger droplet sizes (230 to 355 nm). This change in the plasticizing may be caused by the increasing surface area of interaction in the smaller droplets within the CMC matrix, weakening the CMC-CMC interactions [26]. The same effect was observed in konjac glucomannan films charged with Pickering emulsions; the smallest droplet size (32.18 ± 30.99 μm) resulted in a lower film tensile strength. Elongation at break enhanced with increasing droplet size and reduced again for the biggest droplet size (42.86 ± 15.27 μm). The researchers attributed this behavior to the reorganization of emulsion droplets, as observed by confocal laser scanning microscopy [120]. However, nanosized droplets (39.3 ± 0.7 nm) produced starch films with a higher tensile strength and elongation at break and smoother microstructure than films incorporated with microsized ones (138.1 ± 0.5 nm) [77]. In contrast, tensile strength and elongation at break decreased with the incorporation of Pickering emulsions with larger droplet sizes (140.5 ± 1.450 nm) in chitosan/anthocyanidin films, which may be related to the rougher and looser internal structure [88].

Changes in optical properties, such as transparency and the UV-Vis light barrier, can influence the application of active films. Micrometric droplets of a carnauba-wax emulsion raised the opacity of starch films in relation to nanometric droplets. This can be explained by the higher capacity of larger droplets to disperse light. Moreover, films charged with nanoemulsions were even less opaque than the control film, which is a surprising outcome, according to Oliveira Filho, et al. [77]. With regard to the UV-Vis light barrier, films containing micrometric droplets presented a higher barrier than the nanometric ones, corroborating the opacity results [77]. Similarly, the larger droplets of Pickering emulsions caused the highest opacity and UV-Vis light barrier of chitosan/anthocyanidin films [88]. On the contrary, decreasing droplet sizes of Pickering emulsions reduced the light transmission of konjac glucomannan films due to a greater number of smaller droplets that can promote light blocking or scattering [120].

Regarding the active properties, chitosan/anthocyanidin films charged with the smallest droplets (11.84 ± 0.130 and 13.50 ± 0.240 nm) of Pickering emulsions carrying cinnamon–perilla EOs exhibited the highest antioxidant activity, as assessed by DPPH. The greater stability of particle-stabilized droplets can prevent EO evaporation during emulsion production, preserving the antioxidant activity [88]. Concerning the antibacterial effect against Gram-negative and Gram-positive bacteria, CMC-based films charged with emulsions encapsulating licorice EO with smaller droplets were more effective than an emulsion with bigger droplets. This might be caused by the higher surface-to-volume ratio of the smaller droplets and EO evaporation during the production of the film charged with an emulsion with bigger droplets [26].

### 4.3. Effect of Emulsifier Type

The oil droplets of the emulsions can be coated by different emulsifiers (proteins, carbohydrates, low-molecular-weight surfactants) and solid particles (e.g., nano/microgels from different sources of proteins, cellulose, and chitosan particles). These compounds influence the interfacial layer properties (such as charge, hydrophobicity, surface activity, and thickness) and, therefore, the interaction of the emulsions with the components of the film-forming solutions, resulting in active films with distinct properties [90,95]. There are few works studying the influence of different emulsifiers for emulsion production on film properties.

Using different compounds to stabilize nanoemulsions carrying cinnamon EO, such as ethyl-Nα-lauroyl-L-arginate hydrochloride (LA) alone or co-stabilized by ethyl-Nα-lauroyl-L-arginate hydrochloride and hydroxypropyl-β-cyclodextrin (LH), Xu, et al. [90] observed that WVP was lower for chitosan-based films with the addition of co-stabilized nanoemulsions, which can be associated with their uniform distribution through the chitosan matrix [90].

The mechanical properties were weakened by the incorporation of an LA-stabilized nanoemulsion, resulting in a chitosan-based film with reduced tensile strength and elongation at break. This stabilizer disrupted the crystalline regions of chitosan, causing an interruption of the continuity of the chitosan network structure. However, when LH was used as a co-stabilizer, films presented slightly higher tensile strength and higher crystallinity. Furthermore, the film composed of LH showed a lower elongation at break than the control film but a higher elongation at break than the film incorporated with an LA-stabilized nanoemulsion [90]. Changing the emulsifier also influenced the mechanical properties of gelatin-based films. Films with sodium caseinate-stabilized nanoemulsions encapsulating eugenol showed higher tensile strength, higher stiffness, and lower stretchability than films charged with soy-lecithin-stabilized nanoemulsions. Furthermore, sodium caseinate resulted in a film with lower surface roughness than soy lecithin [95].

The emulsifier type can also affect the active properties of biopolymeric films. Higher antioxidant activity, measured by the ABTS method, was observed in gelatin film incorporated with a sodium caseinate-stabilized nanoemulsion compared with gelatin film with a lecithin-stabilized emulsion. Authors attributed this fact to a better retention of eugenol in oil droplets when sodium caseinate was used as the emulsifier [95]. The antimicrobial effect of active chitosan-based films against *E. coli* was more effective when a nanoemulsion encapsulating cinnamon EO co-stabilized with LH was incorporated into the film-forming matrix. The LH co-stabilizer enhanced the uniformity of EO droplet distribution through the chitosan matrix and better preserved the antibacterial activity of the cinnamon EO. Similar behavior was observed against *S. aureus*, but the antibacterial effect was higher than against *E.coli.* [90].

## 5. Active Film Stability and Bioactive Compound Retention

Some active compounds, like EOs, could evaporate from the film during drying or storage due to its high volatility [124]. Active compounds can be lost by diffusion from the film interior to its surface and by convection from the film surface to the surrounding environment [93]. Therefore, the retention of these compounds in the film matrix and their stability over time need to be evaluated. Evidently, the loss of active compounds implies active films with lower antibacterial and antioxidant activity over time.

Chu, et al. [124] reported significant EO losses during the drying process and during the six days of storage of pullulan-based films loaded with nanoemulsions carrying cinnamon EO. Decreasing the droplet size of the nanoemulsion reduced the loss of the EO, which could be related to the more homogeneous internal structures of these films, as observed by scanning electron microscopy (SEM) and other physical properties. The pullulan matrix could not entrap the high amount of oil droplets, which caused droplet flocculation during film formation, leading to a porous structure and faster release of active compounds [124]. Contrarily to these results, Gahruie, et al. [110] observed that a higher initial concentration of *Zataria multiflora* EOs encapsulated into nanoemulsions incorporated in basil seed gum films decreased the EO loss during storage at room temperature. Nevertheless, they also observed that 50% of the active compound was lost on the first day of storage [110]. Ma, et al. [93] produced chitosan films loaded with emulsified cinnamon bark oil and kept them for 7 days at room temperature. During this period, films incorporated with microemulsion were more efficient in retaining the active compound than the control one [93].

Comparing different emulsifiers in the production of nanoemulsions, Xu, et al. [90] observed that the chitosan matrix charged with a nanoemulsion co-stabilized by LH showed higher retention of cinnamon EO than when only one emulsifier (LA) was used to stabilize the nanoemulsion. Thus, adding LH to co-stabilize the nanoemulsion promoted a better encapsulation of the EO [90]. In another work, the retention rate of cinnamon EO in chitosan films was higher when this bioactive compound was added to conventional emulsions than Pickering emulsions [72]. The authors attributed this fact to the larger droplets of the Pickering emulsion, which caused a bigger pathway for the EO, increasing its volatilization [72].

It is worth noting that the retention capacity of an active compound in a biopolymer matrix during film production and storage generally depends on the matrix structure, composition, and interactions [125]. Furthermore, the emulsion type, droplet size, and concentration of the active compound in the matrix can influence the retention properties and stability of the films, as seen in the examples above.

## 6. Release Properties in Food Simulants

The release process of an active compound from a biopolymeric matrix occurs in three steps. Firstly, the solvent penetrates the matrix, provoking a relaxation and swelling of the biopolymeric chains and facilitating the active compound diffusion to the surrounding medium [112]. A quick release occurs in the first hours, followed by a deceleration in the release rate, and, in the end, an almost constant rate is reached. The fastest release in the first phase can be related to encapsulated compounds in the film surface or close to it. The second phase is associated with the molecules inside the matrix, which show a slow diffusion through the film caused by specific interactions with the biopolymeric chains [126].

For packaging applications, active films will be in contact with food products. Thus, it is essential to understand how active molecules are released and their migration from the film to different foods [127]. For this purpose, release studies have been performed on food simulants [71,105,106,112,115].

Evaluating the release profile of nanoemulsified curcumin from a banana-starch matrix, Sanchez, et al. [19] reported a maximum release value of the simulant for lipophilic foods (50% ethanol *w*/*v*). This fact could be explained by the hydrophobic character of curcumin [19]. Many factors can influence the oil release from a biopolymeric matrix, such as film structure and composition, presence of hydrogen bonds, and solvent type [128].

Zhang, et al. [108] studied the release properties of oregano EO encapsulated in Pickering emulsions from konjac glucomannan films in different food simulants, considering standard food simulants for fatty foods (95% aqueous ethanol), oil-in-water emulsions, alcoholic food (50% aqueous ethanol), and aqueous-based food (distilled water). The authors observed that the release rate of EO reduced with increasing ethanol concentration in the simulant. The solvent could not diffuse quickly into the biopolymeric matrix at the higher ethanol concentration, resulting in a slower release rate. Furthermore, the solvent polarity decreased with the increment in ethanol concentration [129].

Xu, et al. [92] observed a significant decrease in the release of glycerol solution (60%) with an increasing concentration of Pickering emulsions carrying cinnamon EO loaded onto chitosan films. This behavior was attributed to the enhancement in the interaction between the biopolymeric matrix and octenyl succinic anhydride (OSA)-modified gum Arabic, which was used as a stabilizer of the Pickering emulsion [92]. Dammak, et al. [96] reported that the release rate was faster at lower concentrations for nanoemulsions encapsulating rutin from gelatin films.

Shen, et al. [80] evaluated the release rates of nano- and Pickering emulsions encapsulating clove EO loaded onto pullulan–gelatin films in a fatty foodstuff simulant (ethanol 95%). They observed that the EO concentration was insignificant at the release rate; however, emulsion type showed a significant effect. Films incorporated with nanoemulsion had higher release rates than films with Pickering emulsions [80]. Similar behavior was observed by Almasi, et al. [103] in pectin-based films loaded with nanoemulsions and Pickering emulsions encapsulating the marjoram EO at the same conditions. However, EO concentration was significant in this case, and the release was faster at higher concentrations for both emulsions [103]. On the contrary, Hua, et al. [89] reported that the release of clove EO encapsulated in Pickering emulsions in ethanol 95% as a simulant decreased with the increase in nanoparticle concentrations in the chitosan matrix [89]. Chitosan-based films incorporating conventional emulsion encapsulating cinnamon EO presented a slower release than similar films with Pickering emulsion [72].

As can be seen, several factors affect the release of active compounds from films. Some examples include the emulsion type, food simulant, biopolymeric matrix, and interaction between emulsion and matrix. In addition to studying the release in food simulants, verifying the performance of these films in real systems is crucial.

## 7. Applications of Active Films Incorporated with Emulsions

In addition to studying the influence of emulsion incorporation into the biopolymeric matrix, active film performance as food packaging or coating is very important. Therefore, some researchers have applied these films to foodstuffs [88,91,94,99,130,131,132].

Sun, et al. [94] evaluated the effects of gelatin films incorporated with nanoemulsions encapsulating lavender EO to preserve cherry tomatoes. They reported that the active films showed antioxidant activity and antibacterial properties against *E. coli*, *S. aureus,* and *Listeria monocytogenes*, sustained release characteristics, and excellent heat-sealing performance, essential for food-packaging bag production. These active films effectively reduced weight loss, inhibited microorganism growth, and delayed the degradation of titratable acids and phenolic compounds in cherry tomatoes, extending their storage time [94]. Goshal and Shivani [117] applied tamarind starch/whey-protein-concentrate blended films with a nanoemulsion encapsulating thyme EO as tomato packaging. In 14 days of storage, total acidity and total soluble solids were higher in the control fruits, and the active packaging retarded the weight loss in tomato fruits. In sensory evaluation, there were significantly high scores in appearance, firmness, flavor, and overall appearance detected by panelists for packaged tomatoes. Active films could delay the ripening process and provide better-quality to the tomato fruits [117]. It is well known that the quality of fruits can be influenced by the physicochemical properties of the biopolymers used as coatings [132].

Dini, et al. [91] studied an edible coating of a chitosan solution loaded with a nanoemulsion containing the cumin EO. They reported that active-coating combinations extended the shelf life of beef loins by eight days compared with the control group. When active coating was combined with gamma irradiation, the shelf life of beef loins was extended by 15 days under refrigerated storage [91]. Pérez-Córdoba, et al. [131] applied a gelatin/chitosan-based film activated with nanoemulsified garlic EO and α-tocopherol for mortadella conservation. The results showed that the active film was effective against bacterial growth [131]. Yuan, et al. [31] applied a chitosan solution loaded with a double emulsion (W/O/W) carrying nisin in the inner aqueous phase and carvacrol in the oil phase as a coating for salmon fillets. During storage in a refrigerator (4 °C), coated fillets showed the lowest values of pH, water loss, total viable counts, total volatile basic nitrogen, and thiobarbituric acid reactive substance values (TBARS), indicating an increase in the shelf life of the salmon fillets since that these properties are related to the secondary oxidation of lipid products [31]. Zhao, et al. [88] studied the preservation of red sea bream wrapped with anthocyanidin/chitosan films charged with a cinnamon–perilla EO encapsulated in Pickering emulsions. The active films with emulsions and anthocyanidin could maintain the concentration limit of the total volatile basic nitrogen up until 12 days of storage, while the control reached this limit on the 6th day. Furthermore, these packages maintained low TBARS values after 14 days of refrigerated storage, enhancing the shelf life of fish fillets in 6–8 days [88]. Liu, et al. [111] produced a gelatin/chitosan matrix with Pickering emulsion carrying cinnamon EO and free curcumin and applied it for pork meat preservation and monitoring freshness. The film’s color changed according to the meat’s pH variation due to the presence of curcumin. Furthermore, the film containing the Pickering emulsion carrying cinnamon EO showed lower total volatile basic nitrogen than the control film. Thus, this film has great potential to be applied as an active packaging and freshness indicator for pork [111]. Liu, et al. [72] also applied chitosan film charged with a Pickering emulsion encapsulating cinnamon EO for pork meat preservation. The active film retained the freshness of the meat and was capable of maintaining structural integrity during the storage time compared with the control film [72].

These examples demonstrate the potential of biopolymeric active films incorporated with emulsions charged with active compounds to preserve and extend foodstuff’s shelf life. However, more application studies need to be performed in other food systems.

## 8. Final Remarks

Active films based on biopolymers have attracted a lot of attention from researchers in recent years due to their being an eco-friendly material and minimizing the use of chemical additives in food. The focus of the recent studies is the emulsification of lipophilic active compounds, such as EO, because of the difficulties of their incorporation into hydrophilic biopolymeric matrices. However, more recently, some authors have successfully studied the emulsification of some hydrophilic compounds to incorporate into active films.

In addition to favoring the incorporation of non-polar compounds, emulsion-based systems act as protectors of these compounds. A comparison between the addition of the active compound in its natural form and its emulsified form showed a positive effect on the concentration of the active compound, the antioxidant and antimicrobial properties of the films, and their stability during storage.

Active compounds in films are traditional emulsions, nanoemulsions, and Pickering emulsions. This review reports that the emulsion type and droplet size influence film properties, such as mechanical, barrier, and optical properties. In addition, the emulsifier type used to stabilize the emulsion can also affect the film’s properties. The impact of the emulsifier type on the film properties needs to be studied more because, currently, few works are evaluating its influence.

Emulsion characteristics affect the active films’ antimicrobial and antioxidant activities and the active compounds’ release to the surrounding media. Therefore, it is essential to define the application of the active films to select the better emulsion type and production parameters to fabricate the tailored active packaging.

The application of these films in real systems needs to be further studied. This review reports that some applications effectively extended the shelf life of some foodstuffs. However, more studies on the migration of active compounds and their interference with the flavor and aroma of foods must be carried out.

Although biopolymeric matrices incorporated with emulsions encapsulating active compounds present potential to be used as food packaging, there are a few concerns about their commercial scale. These biopolymeric films show a high water vapor permeability rate and high solubility in water. Moreover, the effects of compound migration and the biodegradability of these systems must be studied.

## Figures and Tables

**Figure 1 foods-12-03602-f001:**
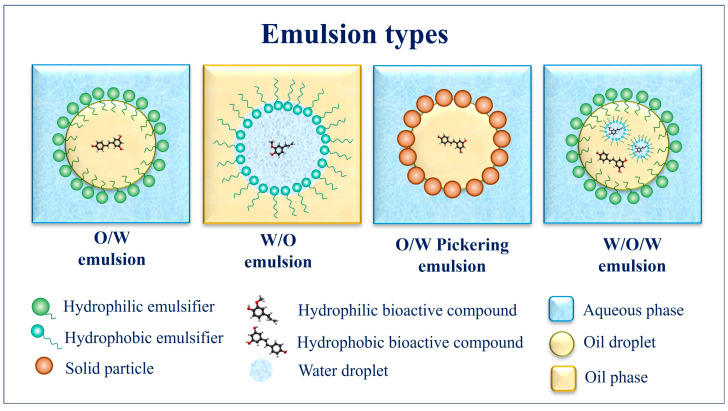
Different emulsion types utilized to encapsulate and protect bioactive compounds to produce biopolymer-based action films. (Source: Authors).

**Figure 2 foods-12-03602-f002:**
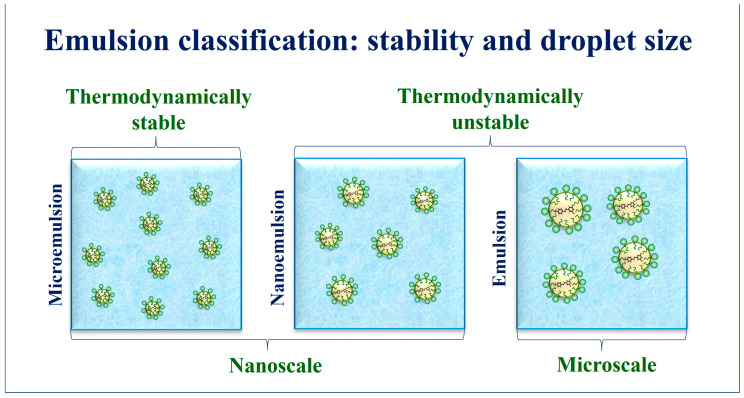
Classification of emulsions according to thermodynamic stability and droplet size. (Source: Authors).

**Figure 3 foods-12-03602-f003:**
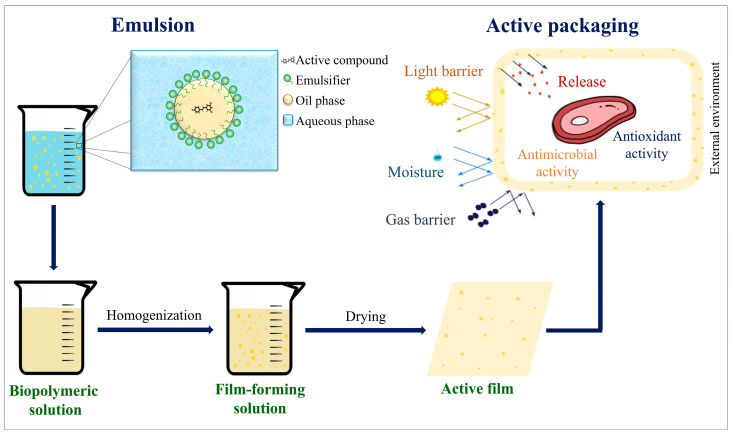
Scheme of the production of active packaging incorporated with emulsified systems and application simulation. (Source: [80] with modifications).

**Figure 4 foods-12-03602-f004:**
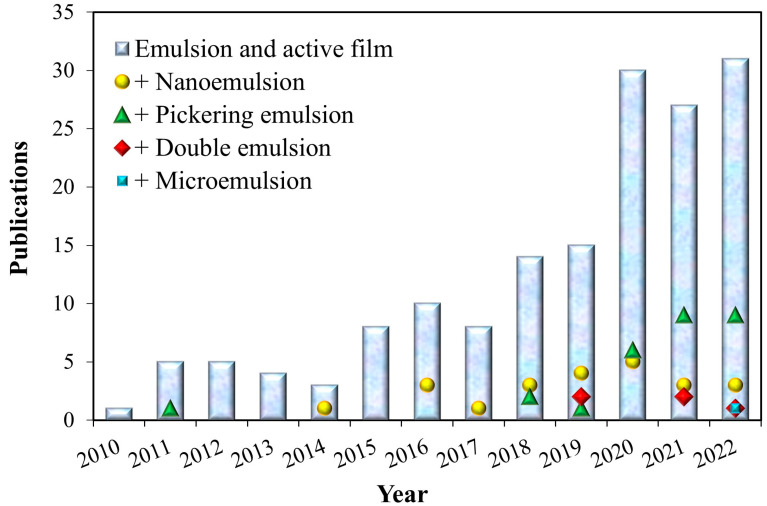
Number of publications using the terms “emulsion” and “active film” in combination with the keywords “nanoemulsion”, “microemulsion”, “Pickering emulsion” or “double emulsion” in the “Food Science Technology” category of the Web of Science Core Collection. (Source: Authors).

**Figure 5 foods-12-03602-f005:**
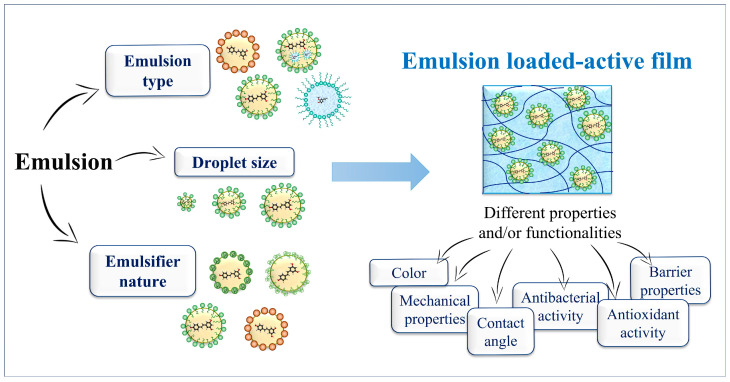
Characteristics of emulsified systems (emulsion type, droplet size, and emulsifier nature) and their effects on the structure and properties of emulsion-loaded active films. (Source: Authors).

**Table 1 foods-12-03602-t001:** Summary of some of the most recent studies that used different emulsion types to encapsulate a range of bioactive compounds before their incorporation into film-forming solutions (source: authors).

Biopolymer	Emulsion Type	Active Compound	References
Chitosan	Pickering	Cinnamon EO	[72]
	Double emulsion	Nisin/carvacrol	[31]
	Pickering	Cinnamon and perilla EO	[88]
	Double emulsion	“Pitanga” leaf extract	[30]
	Pickering	Clove EO	[89]
	Nanoemulsion	Cinnamon EO	[90]
	Nanoemulsion	Cumin EO	[91]
	Pickering	Clove EO	[74]
	Emulsion	Cinnamon EO	[92]
	Microemulsion	Cinnamon bark oil	[93]
	Nanoemulsion	Carvacrol	[18]
Gelatin	Nanoemulsion	Lavender EO	[94]
	Double emulsion	“Pitanga” leaf extract	[30]
	Nanoemulsion	Cinnamaldehyde	[75]
	Pickering	Hesperidin	[17]
	Nanoemulsion	Eugenol	[95]
	Nanoemulsion	α-tocopherol, garlic EO and cinnamaldehyde	[67]
	Nanoemulsion	Rutin	[96]
	Nanoemulsion	Ginger EO	[97]
Starch	Nanoemulsion	Curcumin	[19]
	Pickering	Ho wood, cardamom, and cinnamon EO	[98]
	Emulsion	Lemongrass EO	[76]
	Micro and nanoemulsion	Carnauba wax	[77]
	Pickering	Cinnamon EO	[99]
Whey protein (WPI)	Nanoemulsion	Bergamot oil	[23]
	Nanoemulsion	α-tocopherol	[100]
	Nanoemulsion	*Grammosciadium ptrocarpum* Bioss. EO	[101]
Pectin	Emulsion	Clove EO	[102]
	Nanoemulsion and Pickering	Marjoram EO	[103]
	Nanoemulsion	Copaiba oil	[68]
	Pickering	Cinnamaldehyde	[104]
Sodium caseinate	Nanoemulsion	Cinnamon EO	[105]
Sodium alginate	Emulsion	Cinamon EO	[106]
Soy protein isolate	Micro- and nanoemulsion	Carvacrol and cinnamaldehyde	[51]
Carboxymethyl cellulose (CMC)	Emulsion	Licorice EO	[26]
Cellulose nanofibrils	Pickering	Oregano EO	[107]
Konjac glucomannan	Pickering	Oregano EO	[108]
Pullulan	Nanoemulsion	Cinnamon EO	[109]
	Emulsion	Cinnamaldehyde, thymol and eugenol	[71]
Basil seed gum	Nanoemulsion	*Zataria multiflora* EO	[110]
Chitosan/gelatin	Pickering	Cinnamon EO and curcumin	[111]
	Double emulsion	“Pitanga” leaf extract	[30]
	Nanoemulsion	α-tocopherol, cinnamaldehyde and garlic EO	[28]
	Nanoemulsion	α-tocopherol, garlic EO and cinnamaldehyde	[67]
Chitosan/sodium caseinate blend	Nanoemulsion	α-tocopherol, garlic EO and cinnamaldehyde	[67]
Chitosan/polylactic acid bilayer	Pickering	Thymol	[112]
Gelatin/agar	Pickering	Clove EO	[73,113]
Gelatin/pullulan	Nanoemulsion and Pickering	Clove EO	[80]
Starch/polyvinyl alcohol (PVA)	Pickering	Curcumin	[22]
	Nanoemulsion	Carvacrol	[114]
	Nanoemulsion	Carvacrol	[27]
Pectin/papaya puree	Nanoemulsion	Cinnamaldehyde	[82]
Pullulan/xanthan gum/locust bean gum	Nanoemulsion	Cinnamaldehyde, thymol and eugenol	[115]
Carrageenan/agar	Pickering	Tea tree oil	[116]
Tamarind starch/WPI	Nanoemulsion	Thyme EO	[117]
Konjac glucomannan/pullulan	Pickering	Tea tree EO	[118]

**Table 2 foods-12-03602-t002:** Influence of emulsion characteristics on biopolymeric film properties compared with a control film. (Source: Authors).

Property	Emulsion Type			Droplet Size		Emulsifier	Refs.
	Emulsion	Nanoemulsion	Pickering	Smaller	Bigger		
WVP	Decreased	-	Increased	-	-	-	[72]
	-	Decreased	Decreased	-	-	-	[80]
	-	Decreased	Decreased	-	-	-	[103]
	-	-	-	Decreased	Increased	-	[26]
	-	-	-	Decreased ^2^	Decreased	-	[77]
	-	-	-	Decreased	Increased	-	[88]
	-	-	-	Decreased	Decreased ^2^	-	[120]
	-	-	-			Decreased ^3^	[90]
Moisture absorption/ content	-	Increased	Decreased	-	-	-	[80]
	-	-	-	No effect	No effect	-	[77]
	-	-	-	No effect	Decreased ^2^	-	[120]
	-	Increased	Decreased	-	-	-	[103]
Water contact angle	Increased	-	Increased	-	-	-	[72]
	-	-	-	Increased	Increased	-	[120]
	-	-	-	No effect	No effect	-	[77]
Surface roughness	-	Decreased	Increased	-	-	-	[80]
	-	-	-	No effect	Increased	-	[26]
	-	-	-	Increased	Decreased ^2^	-	[120]
Tensile strength	Decreased	-	Decreased	-	-	-	[72]
	-	Decreased	Increased	-	-	-	[80]
	-	Decreased	Increased	-	-	-	[103]
	-	-	-	Decreased ^2^	Decreased	-	[26]
	-	-	-	No effect	Decreased	-	[88]
	-	-	-	Decreased ^2^	Decreased	-	[120]
	-	-	-	Decreased ^2^	Decreased	-	[77]
	-	-	-	-	-	Increased ^3^	[90]
	-	-	-	-	-	Increased ^4^	[95]
Elongation at break	Decreased	-	Decreased	-	-	-	[72]
	-	Increased	Decreased	-	-	-	[80]
	-	No effect	Increased/decreased *	-	-	-	[103]
	-	-	-	Increased	Increased ^2^	-	[26]
	-	-	-	Increased	Decreased	-	[88]
Elongation at break	-	-	-	Increased	Increased	-	[77]
	-	-	-	-	-	Decreased ^3^	[90]
	-	-	-	-	-	Decreased ^4^	[95]
UV-Vis light barrier	Decreased	-	Decreased	-	-	-	[72]
	-	-	-	Increased	Increased ^2^	-	[77]
	-	-	-	Increased	Increased ^2^	-	[88]
	-	-	-	Increased ^2^	Increased	-	[120]
Antioxidant activity ^1^	-	Highest	Lowest	-	-	-	[80]
	-	Highest	Lowest	-	-	-	[103]
	-	-	-	Highest	Lowest	-	[88]
	-	-	-	-	-	Highest ^4^	[95]
Antimicrobial activity ^1^	No effect	-	Effect	-	-	-	[72]
		No effect	No effect	-	-	-	[103]
	-	-	-	Higher effect	Lower effect	-	[26]
	-	-	-	-	-	Higher effect ^3^	[90]

* Depending on emulsion concentration; ^1^ In comparison with film with emulsion; ^2^ In comparison with smaller droplet addition; ^3^ LH addition; ^4^ Sodium caseinate.

## Data Availability

The data obtained in this study are available from the corresponding author upon reasonable request.

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
