# Peer review of "Films Based on Biopolymers Incorporated with Active Compounds Encapsulated in Emulsions: Properties and Potential Applications—A Review"

_foods, 2023, doi:10.3390/foods12193602_

Round 1
Reviewer 1 Report
In this study, the biopolymers-based emulsified films incorporating active compounds were reviewed. In general, the article is well written. The referee has some remarks that should be added to enhance the quality of the paper:
1) English style should be revised due to the grammatical errors
2) Please add a section to explain the advantages/disadvantages of emulsion-based films compared to other lipidic carriers such as SLN, liposome, NLC, structured oil, nanoparticles, etc.
3) Please add some references in section 4.1 that compare the advantages/disadvantages of single emulsion versus double emulsion in biopolymeric film
4) I recommend adding the main results of each publication to Table 1 in brief.
5) References style in the manuscript should be checked, such as line 261.
English style should be revised due to the grammatical errors
Author Response
We would like to thank you for your suggestions and questions to the paper entitled “Biopolymer-based films incorporated with active compounds encapsulated in emulsions: Properties and potential applications – A review”, which will certainly improve the quality of our article. All of them were considered, so we provided a point-to-point list to help you find them in the text.

Reviewer 2 Report
The manuscript entitled “Films based on biopolymers incorporated with active compounds encapsulated in emulsions: Properties and potential applications – A review” is a review of films incorporated with active compounds, especially essential oil encapsulated in different types of emulsion, e.g., nanoemulsion, Pickering emulsion. The authors described obvious facts about emulsion and different factors affecting film incorporated with active compounds. However, in the Abstract and introduction, the authors suggest that the work will focus on new films used in the food industry, but there was very little information about this application in the work. Additionally, the Abstract suggests that the paper will concern problems and their potential solutions when creating films with the addition of lipophilic compounds. Still, in most cases, only the results obtained by other authors using different solutions without delving into explanations are given. In recent years, many papers have focused on films based on biopolymers incorporated with active compounds encapsulated in emulsions. Therefore, the manuscript’s significance, originality, and contribution to new knowledge in the field is controversial. The comments that follow are listed, not by importance, but by order of appearance:
Lines 71-75 The goal of papers should be more precise. It should be specified what will be described in subsequent chapters as in the previous work: DOI: 10.1039/d1ra04888k
Chapter 2 Emulsion
There is a lot of obvious information about emulsion there. Maybe it will be good to add some detailed information about emulsion stabilization.
Chapter 3. Films incorporated with bioactive compound-charged emulsions
Lines 145-155 - One production method is described. It may be worth describing more options for receiving films and their advantages and disadvantages.
Line 198; 262; 301 – essential oil was replaced with the abbreviation EO on line 196. Therefore, use an abbreviation throughout the text
Table 1. Summarizes some of the most recent studies that used different emulsion types to encapsulate a range bioactive compound before their incorporation into film-forming solutions.
I suspect that the font in the table is too large compared to the rest of the text. Additionally, the information in the table could be expanded to include the possibility of using the mentioned films in the food industry, which, according to the introduction, is the article's subject.
The polymer mixture is sometimes written in upper case letters and sometimes in lower case letters. Please standardize this.
Additionally, in the table:
Chitosan/gelatin Nanoemulsion a-tocopherol, cin- namaldehyde and garlic EO – please connect cinnamaldehyde
Chitosan/gelatin Nanoemulsion a-tocopherol, cin- namaldehyde and garlic EO – should be α-tocopherol
Gelatin/agar is the same; please connect citation [70, 121]
PVA abbreviation is not explained
Lines 218-232- Why don't you mention an additional, crucial function of this film, which is the pH indicator
Line 238 – There is too much space between which form
Line 261 – J. Liu, Song, et al. should be replaced by Liu, et al.
Line 277-295 The authors only describe the results; it may be worth also describing the explanation of the phenomenon, e.g., in citation 79, they are given.
Line 343 – It should be emphasized that this concerns gelatin film with W/O/W double emulsion.
Line 364-369 - The reference should be added.
Line 378-388 - In reference 119, the droplet size and other parameters are connected with the amount of oil phase(10%, 30%, 50% and 70%, v/v).
Line 388 abbreviation CMC was used before the explanation should be used after using the first time.
Line 396 – It should be Liu et al.
Line 419 – The authors should be written
Table 2
Table 2 placing the table in point 2 of section 4.3. The effect of emulsifier type is incorrect. Please place it next to the text describing these properties. This table should be on one page.
The emulsion type should be separated from the droplet size
Are you sure these parameters depend only on the type of emulsion, droplet size, and emulsifier? Does the use of different biopolymers or different emulsion concentrations affect these parameters? This table seems incomplete without information about the composition of the film.
Line 506 – nanocmulsion it should be nanoemulsion
Line 547 – The abbreviation OSA is not explained
7. 7. Applications of active films incorporated with emulsions. This chapter should be divided into different types of food, e.g., meat, dairy, vegetable, and fruit.
Line 587 – The abbreviation TBARS is not explained. Additionally, there should be information about connection of TBARS with secondary oxidation products.
Line 595 – It should use the abbreviation TBARS.
Line 630 - Common emulsion is a colloquialism that should not be used in scientific work
Author Response

(The authors gave the same response as above.)

Reviewer 3 Report
Please find below the specific comments:
Films based on biopolymers incorporated with active compounds encapsulated in emulsions: Properties and potential applications – A review
Abstract
OK
Keywords
OK
Introduction section
This section is complete, authors consider key information.
Emulsions
This section is complete
Films incorporated with bioactive compound-charged emulsions
Please improve the Figure 3, some colors of the words are fuzzy and complicated to read, also modify the pixels for more quality
For Figure 4 improve the pixels for better quality
Figure 5 OK
Table 1 OK
Table 2 OK
Active films stability and bioactive compound retention
OK
Release properties in foods simulants
OK
Application of active films incorporated with emulsions
OK
Final remarks
OK
Please consider adding the following manuscripts, you can find valuable information:
§ 10.1016/j.foodcont.2022.109063
Author Response
We would like to thank you for your suggestions and questions to the paper entitled “Films based on biopolymers incorporated with active com-pounds encapsulated in emulsions: Properties and potential applications – A review”, which will certainly improve the quality of our article. All of them were considered, so we provided a point-to-point list to help you find them in the text.

Round 2
Reviewer 2 Report
Dear Authors,
I have gone through the revised version. It seems that the manuscript was greatly improved and therefore, it can be recommended for publication in Foods. However, before printing, make sure that the chapter titles are not at the end of the page (see line 471)
Best regards